# Respectfulness-processing revisited: An ERP study of Chinese sentence reading

Liyan Ji[1,2]*, Lin Cai[3], Aiai Ji[4]

**1** School of Psychology, Fujian Normal University, Fuzhou, China, **2** School of Psychological and Cognitive Sciences, Peking University, Beijing, China, **3** Department of Electronics and Electrical Engineering, Keio University, Yokohama, Japan, **4** Mental Health Education Center, Tiangong University, Tianjin, China

* jiliyan0717@126.com

## Abstract

In Mandarin Chinese, an important manifestation of respectfulness is the use of different forms of second-person pronouns. Jiang et al. (2013) examined the event-related potential (ERP) correlates of processing respectful and plain pronouns in Chinese. However, this study suffered from a few methodological limitations, which restricted both the reliability and functional interpretations of the study's findings. In the present study, we resolved these limitations and further investigated the neurocognitive mechanisms of processing the respectfulness of pronouns. In the present study, participants read 160 critical Chinese sentences with a second-person pronoun (*ni* or *nin*) that was either consistent or inconsistent with its prior sentence context in terms of respectfulness, as well as 240 filler sentences. Unlike the previous study that reported a 300–500 ms negative response (N400) for both types of inconsistent pronouns, a sustained positive response for Nin inconsistent and a sustained negativity response for Ni inconsistent in the late time window, the present study found an N400 response and late sustained negativity for Nin inconsistent, but not for Ni inconsistent. Furthermore, the cluster-based permutation showed a significant negative cluster for Nin inconsistent, extending from 432–622 ms. We related this negative response for Nin inconsistent with recent accounts of the N400 and late negativity. Finally, the absence of the ERP effect for the Ni condition was linked to the role of the pragmatic property of Ni.

## Introduction

Politeness is a significant feature of linguistic communication. Typically, people adjust their language according to the social status of the addressees to show politeness. Many languages, such as Japanese, Korean, and Spanish, use a very complex system of politeness reflected through language. One way to convey social status information during interpersonal communication is to use honorific forms [1].

Previous studies have investigated the underlying neural mechanism of verbal politeness expressed with honorific markers in Japanese and Korean [2–6]. For example, there are two honorific forms in Japanese: the exalted and humble verbs are used to represent respect for the person in the subject and object, respectively. In a recent study [6], Tokimoto et al. investigated

**Data Availability Statement:** The data underlying the results presented in the study are available from OSF database (https://osf.io/zf35p).

**Funding:** This work was supported by the Natural Science Foundation of China (grant number:

31371021), the Wenghongwu Original Research
Funding (WHW20180).

the neural substrate of the processing of Japanese honorification by visually presenting appropriate and anomalous sentences containing the two types of verbs and analyzing the electroencephalogram (EEG) elicited by them. The results showed an N400 response for anomalous verbs compared with the appropriate verbs for the person in the subject and a frontocentral distributed N400 response and a P600 response for the anomalous verbs for the person in the object compared to the appropriate verbs. However, the politeness expressions differ in various languages. In Mandarin Chinese, an important manifestation of politeness is the use of different forms of singular second-person pronouns. However, the neural correlates associated with politeness processing in Mandarin Chinese are unclear.

In Mandarin Chinese, the status of the addressee can be reflected in the attitude of respectfulness showed by the second-person singular pronoun *nin* ("you $_{[respectful]}$") or *ni* ("you $_{[plain]}$"). When a speaker of lower status talks to an addressee of higher status, they are expected to use the respectful form *nin* to show respect toward the addressee. Meanwhile, a speaker of higher status talking to an addressee of lower status is expected to use the less respectful form *ni* to show social dominance or implement a command. For example, a sentence like "The boy said to his grandfather: 'I am very grateful for your$_{[plain]}$ care.'" is accurate at the semantic and syntactic levels. However, this is a violation of politeness. How is this violation detected by the brain? Does it differ from semantic and general world-knowledge violations? Thus, the present study aims to investigate the neural basis of respectfulness processing in Mandarin Chinese.

Only one study conducted by Jiang et al. [7] has examined the event-related potential (ERP) correlates of the processing of respectful and plain forms of second-person pronouns in Chinese. In their study, Jiang et al. found that with respect to the use of the pronoun *nin*$_{[respectful]}$, the status inconsistent condition (e.g., Professor Zhang said to student Lin: "your$_{[respectful]}$ . . .") elicited anterior negativity at 300–500 ms and a late sustained positivity at 800–1600 ms window relative to the status consistent condition (e.g., Student Lin said to professor Zhang: "your$_{[respectful]}$ . . ."). In contrast, when compared to the *ni*$_{[plain]}$ status consistent condition (e.g., Professor Zhang said to student Lin: "your$_{[plain]}$. . ."), the status inconsistent condition (e.g., Student Lin said to professor Zhang: "your$_{[plain]}$ . . .") elicited broad negativity at 300–500 ms and a late sustained negativity effect in the 500–1600 ms window. The authors claimed that the N400 effect reflects the process of integrating semantic information into the context, with the late positivity expressing a non-literal (sarcastic) interpretation of the anomalous *nin*$_{[respectful]}$, and the late negativity reflecting revisions for the anomalous *ni*$_{[plain]}$. However, there are a few methodological limitations in their study, which may restrict the theoretical implications of the ERP correlates observed by them.

## Critique of Jiang et al.'s study

First, the necessary fillers are lacking, and all anomalies occur on the pronouns. In Jiang et al.'s study [7], all fillers are of the same structure: each experimental list has 270 critical sentences with 45 for each condition and 90 fillers of the same structure, including 45 each starting with first (wo-de, "my") and third-person pronouns (ta-de, "his" or "her"), respectively. The fillers are all acceptable; thus, all unacceptable sentences occur in the critical sentence materials using the second-person pronouns. Furthermore, only critical sentences starting with position names allow readers to anticipate the pronoun from the presence of the speaker. In this way, participants understand the purpose of the study and form specific response strategies. Second, the experimental materials used by these authors include some inappropriate possessum objects in the epoch analyses. For example, the use of the word "future" in the sentence, "The grandson said to the grandfather: 'I am worried about your *future*.'" which a junior would not usually say to an old person. Contrastingly, elders often say "future" to the junior; thus, this

situation cannot be balanced among the four conditions. According to previous studies [8–10], compared to high expectation words, low expectation words induced more negative ERP responses. Consequently, the ERP response observed at critical target words is confounded by the effect of different possessum-object appropriateness across conditions. Third, the potential baseline problem (spillover effect) needs to be addressed: the effects elicited by the prior word, having a sustained time course into the post-target onset latency ranges [11]. In Jiang et al. [7] 's study, although the word before the critical word was the same across conditions, the potential baseline difference was likely due to the entire initial context, and the context among different conditions was unbalanced. This difference can spill over to the critical region. Thus, the N400 effect could be due to the baseline difference since they did not examine whether there were baseline differences.

## Rationale and results expectation of this study

In the present study, first, we used various fillers, including semantic and syntax anomalies, and ensured that items with and without violations were equal in number. Second, we modified the critical sentences in the study by Jiang et al. [7]. A half-sentence (only including the interlocutors and the possessum-object; the second person pronoun is consistent with the social status relation of the interlocutors. For example, Professor Zhang said to Student Lin: "your$_{[plain]}$ article. . .") appropriateness rating test was conducted before the ERP experiment. We removed sentences with possessum-objects with a below-average score. Furthermore, only the critical word "your" was included in the epoch in the ERP analysis in the present study. Third, to determine whether contextual differences upstream of the critical word infect the target word's ERPs, we examined the ERP response of pre-target words (time lock to the first word) using the blank interval before the presentation of the first word as a baseline (see Methods for more details).

Among the growing number of linguistically relevant ERP components, three seem to be most frequently reported in pragmatic studies: the N400, the late positivity component, and the late negative component. As mentioned above, the N400 is a well-established ERP component that is generally linked to context-based upcoming word expectations [12]. The processing of pragmatic inference also influences late ERP components. The late positivity component (LPC/P600) has been reported in studies on irony [13,14]. This component has been related to reflecting the truly pragmatic interpretive processes needed to derive the speaker's intention. However, some studies have reported that the late positive ERP response is task-relevant [15,16]. Schacht et al. [16] investigated whether the P600 component to the syntactic violation would be affected by changing the task from correctness judgment to probe verification task. The results showed that P600 to syntactic violation was absent when using a probe verification task; instead, late frontal negativity was elicited. This indicates that the late positive activity reflects error monitoring, at least to some extent, influenced by participants' processing strategies. The late negative component or sustained frontal negativity could also be observed in unexpected words that create an extralinguistic meaning [17,18] or a difficulty in understanding [19]. For example, Bambini et al. [17] investigated the neural basis of metaphor processing in a literary context. Their results showed that the critical word in the metaphorical condition elicited a larger N400, followed by sustained frontal negativity compared to the word in literal expressions. Further analysis using linear mixed models—which can take both by-item and by-participant random variation into account—showed that the late negativity effect is modulated by familiarity, with a more negative response for fewer familiar metaphors. Late negativity has been interpreted as a reflecting mechanism related to the drawing of an array of weak implicatures.

By using various fillers and a novel set of critical sentences as well as checking potential baseline problems, we set out to resolve the limitations of Jiang et al's study and further investigate the neurocognitive mechanisms underlying the respectfulness processing of pronouns during language comprehension. In Jiang et al.'s study [7], they observed a negative response in the time window of 300 to 500 ms (N400) for both types of inconsistent pronouns, even a stronger and broader distribution of the N400 effect for $ni_{[plain]}$. This result is opposite to our experience since $ni_{[plain]}$ can be used in scenarios where there is a close relationship between a speaker of a lower status and an addressee of higher status (e.g., between a daughter and her mother). However, $nin_{[respectful]}$ is typically used by people more carefully than the previously mentioned application. Therefore, we hypothesized that when a speaker of a higher social status (e.g., mother) used the respectful form "$nin_{[respectful]}$" to an addressee of a lower social status (e.g., daughter), it would induce difficulty in understanding. In the current study, we expected to observe the N400 effect and perhaps a late negative activity for the *Nin* inconsistent condition, but not for *Ni* inconsistent condition. As mentioned above, late positivity is task-dependent, which may reflect strategy processing. In Jiang et al.'s study [7], every trial is followed by a comprehension question, and all anomalies occur on the second person pronoun. Under such circumstances, participants may adopt a specific strategy (e.g., error monitoring or guessing what question would be asked), which can increase the effects on the late component. In the present study, only 25% of the trials were randomly selected following a comprehension question task to ensure that participants read the sentences attentively, and the experimental environment was enriched to minimize the strategy processing. Therefore, we did not anticipate the presence of late positivity for any condition.

## Method

### Participants

A total of 43 undergraduate and graduate students provided informed written consent to participate in the experiment. All participants were native Mandarin speakers who were born and raised in Beijing, with no experience of living elsewhere for more than three months. They reported no reading disabilities or known neurological disorders and were all right-handed with normal or corrected-to-normal vision. Data from 11 participants were discarded due to excessive artifacts in the ERP data. The remaining 32 were entered into the analyses (age range, 19–26 years; mean, 21.8 ± 2.53 years; 16 women). The study protocol was approved by the Ethics Committee of the School of Psychological and Cognitive Sciences, Peking University. The study was conducted in accordance with the ethical principles stated in the Declaration of Helsinki.

### Design and materials

We used a full factorial design that crossed two within-subject factors: consistency between the status of the interlocutor and the form of the second person pronoun (consistent or inconsistent) and pronoun type (*Nin* or *Ni*), resulting in four types of sentences (Table 1). The critical word was the different form of the second-person pronoun (nin, ni). There is a high-frequency usage for both nin (515 per million) and ni (39629 per million) [20].

Critical materials comprised 160 sets of sentences containing direct speech. The critical materials were assigned to four lists using a Latin square procedure. In each list, 160 critical items (40 for each type of sentence) were pseudo-randomly mixed with 240 filler sentences. Of the filler items, 80 had almost the same sentence structure as critical items. However, the direct speech began with a first- (*wo-de*, "*my*") or third-person possessive pronoun (*ta-de*, "*his*" or "*her*") to prevent participants from predicting the appearance of a second-person pronoun

**Table 1. Design and stimulus examples for all four critical conditions are given in Chinese, with English glosses and translations.**

| Sentence type | Example |
|---|---|
| **Nin Consistent** | 林同学/对/张教授/说: "/您的/论文/我/已经/收到了。"/ |
| | **Gloss**: Student Lin/ to/ Professor Zhang/ said: "/ nin-de, 'your [respectful]'/ article/ I/ have received." |
| | **Translation**: Student Lin said to Professor Zhang that I have received your article. |
| **Nin Inconsistent** | 张教授/对/林同学/说: "/您的/论文/我/已经/收到了。" |
| | **Gloss**: Professor Zhang/ to/ Student Lin / said: "/ nin-de, 'your [respectful]'/ article/ I/ have received." |
| | **Translation**: Professor Zhang said to Student Lin that I have received your article. |
| **Ni Consistent** | 张教授/对/林同学/说: "/你的/论文/我/已经/收到了。" |
| | **Gloss**: Professor Zhang / to/ Student Lin / said: "/ ni-de, 'your [plain]'/ article/ I/ have received." |
| | **Translation**: Professor Zhang said to Student Lin that I have received your article. |
| **Ni Inconsistent** | 林同学/对/张教授/说: "/你的/论文/我/已经/收到了。" |
| | **Gloss**: Student Lin / to/ Professor Zhang / said: "/ ni-de, 'your [plain]'/ article/ I/ have received." |
| | **Translation**: Student Lin said to Professor Zhang that I have received your article. |

The critical words were underlined.

based on their reading of the sentence context before the direct speech. The remaining 160 filler sentences had other sentence constructions (none containing direct speech) and consisted of 80 correct and incorrect sentences each. In half of these incorrect filler items, the subject or object noun phrase contained syntactic and semantic incoherence caused by the insertion of a degree adverb immediately before the subject or object noun. The other half contained a semantic incoherence that occurred either between an adjective and a noun or between the main verb and its object. For the 80 correct filler sentences, half of them had the same sentence structure as those containing semantic incoherence and/or syntactic incoherence sentences. The other half included an active marker ba or passive marker bei (examples of filler sentences can be found in Supplementary S1). While in Jiang et al. [7] each problematic sentence contained an anomaly on the second-person pronoun, in the current study, the problematic sentences contained various anomalies (syntactic, semantic, or pragmatic).

## Procedure

Stimuli were presented and the responses were recorded using Presentation software (http://nbs.neuro-bs.com/). Participants were seated in a dimly lit and sound-attenuated room. Each participant was instructed to ensure minimal movement of the head or body and to keep their eyes fixated on a cross at the center of the computer screen before the onset of each scenario. At the start of each trial, a white fixation cross appeared at the center of a black screen for 800 ms, followed by a 400 ms blank screen. Sentences were presented in white characters on a black background, segment-by-segment (word or short phrase) in a rapid sequence in the center of the screen. The critical sentences consisted of a series of nine frames, with four segments each for the conversational context and utterance; the fillers included seven to nine frames. Each segment was presented for 400 ms, followed by a blank screen for 400 ms, and then another segment, until the final segment of the sentence. The screen was then left blank for 1300 ms, following which a verification sentence (either a paraphrase of a sentence within the discourse or a paraphrase sentence of information conflict with the critical sentence) was presented 25% of the time. Participants were asked to decide whether the verification sentence correctly expressed the content of the preceding discourse by using one of the two response buttons (the left and right buttons on the computer mouse) under the left and right thumbs. The number of consistent and inconsistent probes was equal for each condition. Each probe

sentence remained on the screen until the participant made a "yes" or "no" response or for a maximum of 3000 ms. The next trial began after a 1000 ms blank interval.

Participants were randomly assigned to one of the four experimental lists. The 400 sentences in each list were divided into five blocks of 80 sentences. For each list, trials were pseudo-randomized so that no more than three consecutive sentences contained the same condition, and at least one verification sentence appeared within five consecutive trials. At the beginning of each block, at least four fillers were presented in succession to warm up the participants. Before the experimental blocks, participants received a practice block of 24 trials, which had the same composition as the critical stimuli. The experimental session lasted for around two hours.

## EEG recordings

The EEGs were recorded from 64 Ag/AgCl electrodes in a secured elastic cap (EASYCAP GmbH, Germany). The EEGs were referenced online to the nose tip. The vertical electrooculogram was monitored from an electrode located above the right eye, and a horizontal electrooculogram was placed at the outer canthus of the right eye. The electrode impedances were kept below 5 kΩ. EEG signals were sampled at 500 Hz with a band-pass filtered 0.016–100 Hz online and were filtered again offline with a band-pass of 0.1–40 Hz for data analysis.

## EEG data preprocessing

Although the word before the critical word was the same across the four conditions, the context (relative social status of interlocutors was different among the four conditions) between consistent and inconsistent conditions could involve the processing of unbalanced reasoning. To rule out this potential impact of the pre-stimulus difference on the ERP effect for the critical word, we checked baseline differences by using blank intervals before the presentation of the first word. A long epoch was computed from 200 ms before the onset of the first word to 3200 ms after the onset of the first word. Then the time window of 3000–3200 ms was the 200-ms before presenting the critical word. Statistical analyses were conducted on the average amplitude during 3000–3200 ms. The results showed that there was no significant main effect or interaction involving factors of pronoun type or pronoun consistency ($ps > 0.1$; see Table 2).

**Table 2. The overall analyses of variance for three time windows (in milliseconds) (N = 32).**

| Source | df | 300–500 ms | | | 500–800 ms | | | Baseline 3000–3200 | | |
|---|---|---|---|---|---|---|---|---|---|---|
| | | F | p | MSE | F | p | MSE | F | p | MSE |
| Con | 1, 31 | 3.01 | 0.093 | 19.45 | 9.47 | 0.004 | 18.53 | 2.01 | 0.17 | 83.23 |
| Pro | 1, 31 | 8.63 | 0.006 | 28.20 | 2.28 | 0.14 | 31.58 | < 1 | | |
| Con × region | 4, 124 | < 1 | | | 1.37 | 0.25 | 1.16 | 1.04 | 0.38 | 6.23 |
| Con × hem | 2, 62 | 1.08 | 0.35 | 2.48 | 2.87 | 0.07 | 1.47 | 1.00 | 0.36 | 10.43 |
| Con × region × hem | 8, 248 | < 1 | | | 1.16 | 0.33 | 0.43 | < 1 | | |
| Pro × region | 4, 124 | 4.32 | 0.032 | 2.75 | 1.81 | 0.18 | 1.86 | < 1 | | |
| Pro × hem | 2, 62 | 11.52 | 0.002 | 1.86 | 2.20 | 0.12 | 1.71 | 1.52 | 0.23 | 6.90 |
| Pro × region × hem | 8, 248 | 1.65 | 0.15 | 0.34 | 1.07 | 0.39 | 0.45 | < 1 | | |
| Con × Pro | 1, 31 | 6.09 | 0.019 | 22.65 | 1.92 | 0.18 | 23.84 | < 1 | | |
| Con × Pro × region | 4, 124 | 3.82 | 0.05 | 3.46 | 1.60 | 0.18 | 2.51 | 1.02 | 0.38 | 5.09 |
| Con × Pro × hem | 2, 62 | < 1 | | | < 1 | | | < 1 | | |
| Con × Pro × region × hem | 8, 248 | 1.06 | 0.39 | 0.39 | < 1 | | | < 1 | | |

Con: Consistency, Pro: Pronoun type, hem: Hemisphere.

Thus, potential baseline differences among the four conditions could be excluded, and baseline correction was conducted in the following analyses. The raw EEG data were first corrected for eye-blink artifacts using the ocular artifact correction algorithm implemented in the BrainVision Analyzer 2.2 software package. The EEG data were then filtered with a band-pass of 0.1–40 Hz. Epochs were extracted from 200 ms before to 1000 ms after the onset of the critical words. Epochs contaminated by a lateral eye movement, voltage drift, or other artifacts were identified with semiautomatic artifact rejection (automatic criterion: amplitudes greater than ± 65 μV, followed by a manual check). The minimal number of trials for each condition was set at 30, and 11 participants were excluded after the artifact rejection. The overall rejection rate for participants included in the final analysis was 9.39%, and a 2 (consistency: consistent *vs*. inconsistent) ×2 (pronoun type: nin *vs*. ni) analysis of variance (ANOVA) showed that the rejection rate was equal for all four conditions (Nin-de consistent, 10.08%; Nin-de inconsistent, 7.58%; Ni-de consistent, 10.23%; Ni-de inconsistent, 9.06%).

## Statistical analysis

We analyzed the EEG data in two ways. We first conducted repeated-measures ANOVAs on the time window of N400 (300–500 ms) and P600 (500–800 ms), as Jiang et al. [7] did intended for a direct comparison of results. A 2 (pronoun type) × 2 (pronoun consistency) × 3 (hemisphere) × 5 (region) repeated measures ANOVA was conducted for the average amplitude of the selected time windows. The chosen topographical factors were the same as in Jiang et al. [7] including a hemisphere with three levels (left, medial, and right) and regions with five levels (frontal, frontocentral, central, centroparietal, and parietal). The region and the hemisphere were crossed, resulting in 15 regions of interest (ROIs), each having three representative electrodes: left frontal (F3, F5, F7), left frontocentral (FC3, FC5, FT7), left central (C3, C5, T7), left centroparietal (CP3, CP5, TP7), left parietal (P3, P5, P7), medial frontal (F1, FZ, F2), medial frontocentral (FC1, FCZ, FC2), medial central (C1, CZ, C2), medial centroparietal (CP1, CPZ, CP2), medial parietal (P1, PZ, P2), right frontal (F4, F6, F8), right frontocentral (FC4, FC6, FT8), right central (C4, C6, T8), right centroparietal (CP4, CP6, TP8), and right parietal (P4, P6, P8). Comparisons were planned for each ROI if the interactions were significant. The Greenhouse-Geisser correction was applied for evaluating the effects with more than one degree of freedom in the numerator. The Bonferroni correction was used for the planned comparisons.

In the second analysis, we used a data-driven approach: cluster-based permutation tests implemented in the Fieldtrip software package [21], and custom-built scripts adopted from Wang and Zhang [22]. This non-parametric test optimally handles the multiple comparisons problem and effectively controls for type I error. With ERP amplitude as the dependent factor, the cluster-based permutation test was performed on each electrode and time point. All t-values neighboring in space and time that cross a predefined threshold ($p < 0.05$) were summed up and grouped as a cluster. For each cluster, the number of significant points was used as the cluster-level test statistic. Next, a null distribution of cluster-level test statistics was created using the Monte Carlo method with 1000 random draws. Finally, the observed cluster-level statistics were compared against the null distribution to assess the significance. This method allowed for the identification of small sustained and large transient effects that might remain undetected by subjectively choosing the time window and electrodes.

To examine the interaction between pronoun type and consistency, cluster-based permutation tests were performed to compare the amplitudes of the two difference-difference waves (Nin inconsistent minus Nin consistent *vs*. Ni inconsistent minus Ni consistent), with significant differences indicating the presence of a two-way interaction. In case this interaction

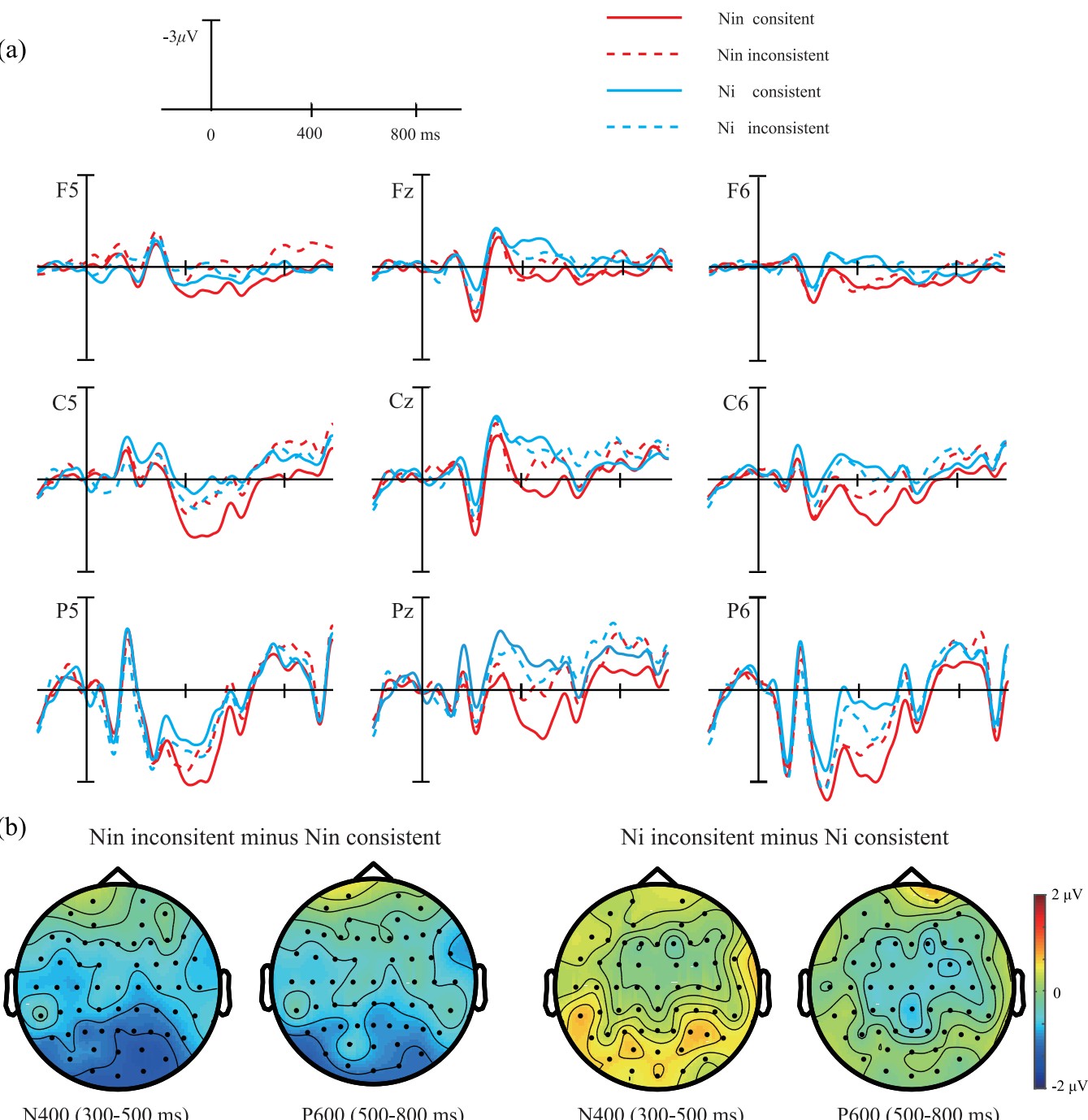

**Fig 1. ERPs time locked to the onset of the critical words for all 32 participants.** (a) Grand average ERPs for all four critical conditions at nine scalp sites; epoch from 200 ms before to 1000 ms after the onset of the critical words. In this figure, the onset of the critical words is at 0 ms, negativity is plotted upwards, and waveforms are filtered (15 Hz low pass, 24 dB/oct). (b) The scalp topographies of the two difference waves in the time windows of N400 and P600.

reached significance, we further examined the simple effects of pronoun consistency at the different levels of pronoun types ("Nin inconsistent *vs*. Nin consistent" and "Ni inconsistent *vs*. Ni consistent"). The grand average ERP waveforms and topographic distributions are shown in Fig 1.

## Results

### Behavioral results

Average accuracy was computed as the percentage of correct responses. The overall average accuracy was 94.38%, suggesting that participants read the sentences attentively. The average accuracy was 94.06 ± 7.98% for Nin consistent, 92.19 ± 6.59% for Nin inconsistent, 95.00 ± 6.22% for Ni consistent, and 95.31 ± 6.21% for Ni inconsistent. An ANOVA with consistency and pronoun type as within-subjects factors revealed a marginally significant main effect of pronoun type ($F$ (1,31) = 4.12, $p$ = 0.05, $MSE$ = 32.03), and the percentage of correct responses for Nin was lower than that for Ni (93.13% vs. 95.16%). Neither was the main effect of consistency significant ($F$(1,31) < 1), nor did it show an interaction between the two factors ($F$ (1,31) < 1). These results indicate that comprehension of sentences with Nin is more difficult than that of Ni.

### Results of the ANOVA analysis

**The 300 to 500 ms time window.**   Repeated-measures ANOVA with pronoun type, consistency, hemisphere, and region as within-subject factors revealed a significant effect of the pronoun type ($F$ (1,31) = 8.62, $p$ = 0.006, $MSE$ = 28.19, $\eta^2$ = 0.22). A two-way interaction between the pronoun type and consistency ($F$ (1,31) = 6.09, $p$ = 0.019, $MSE$ = 22.65, $\eta^2$ = 0.16) indicated that the consistency effect differed between pronoun types. Bonferroni-corrected pairwise comparisons showed that the Nin inconsistent condition (mean = 0.16 μV) elicited a significant larger N400 than the Nin consistent condition (mean = 1.05 μV, $p$ = 0.004). However, there was no significant difference between the Ni consistent (mean = −0.20 μV) and Ni inconsistent condition (mean = -0.01 μV, $p$ > 0.1). A three-way interaction of pronoun type, consistency, and region was significant ($F$ (4,124) = 1.09, $p$ = 0.049, $MSE$ = 3.46, $\eta^2$ = 0.11). Pairwise comparisons showed that the interactions between pronoun type and consistency were significant in all five regions ($ps$ < 0.05). Nin inconsistent elicited a larger N400 response than Nin consistent condition in all five regions, and there was no significant difference between the Ni consistent and Ni inconsistent condition.

**The 500 to 800 ms time window.**   The overall ANOVA revealed a significant effect of consistency ($F$ (1,31) = 9.47, $p$ = 0.004, $MSE$ = 18.52, $\eta^2$ = 0.23). The amplitude was significantly more negative in the inconsistent (mean = -0.40 μV) than in the consistent conditions (mean = 0.21 μV). Although the interaction between the pronoun type and consistency was not significant ($F$ (1,31) = 1.92, $p$ = 0.18, $MSE$ = 23.84), the amplitude difference between Nin inconsistent and Nin consistent (-0.36 vs. 0.56μV) was numerically larger than that between Ni inconsistent and Ni consistent (-0.43 vs. -0.14μV). To further examine these differences, Bonferroni-corrected pairwise comparisons were performed, and the result showed that the difference between Nin inconsistent and consistent was significant ($p$ = 0.04); conversely, the difference between Ni inconsistent and consistent was not significant($p$ > 0.1).

**Results of the cluster-based permutation tests.**   The cluster-based permutation tests revealed a significant interaction between the pronoun type and consistency, extending from 432 to 516 ms (cluster statistic = 2097.3, $p$ = 0.038). Further simple analyses were conducted to compare Nin inconsistent and Nin consistent. A significant negative cluster was detected. F4, F5, FC1, FC2, FC3, FC4, FC5, FC6, FT7, FT8, C1, C2, C3, Cz, C4, C5, C6, T7, T8, CP1, CP2, CP3, CPz, CP4, CP5, CP6, TP7, TP9, P1, P2, P3, Pz, P4, P5, P6, P7, P8, PO3, PO4, POz, PO7, PO8, O1, Oz, and O2 were grouped as a significant negativity cluster for Nin inconsistent compared to Nin consistent condition with the time interval ranging from 432 to 622 ms (cluster statistic = 5442.3, $p$ = 0.004, the significant electrodes have been highlighted in Fig 2).

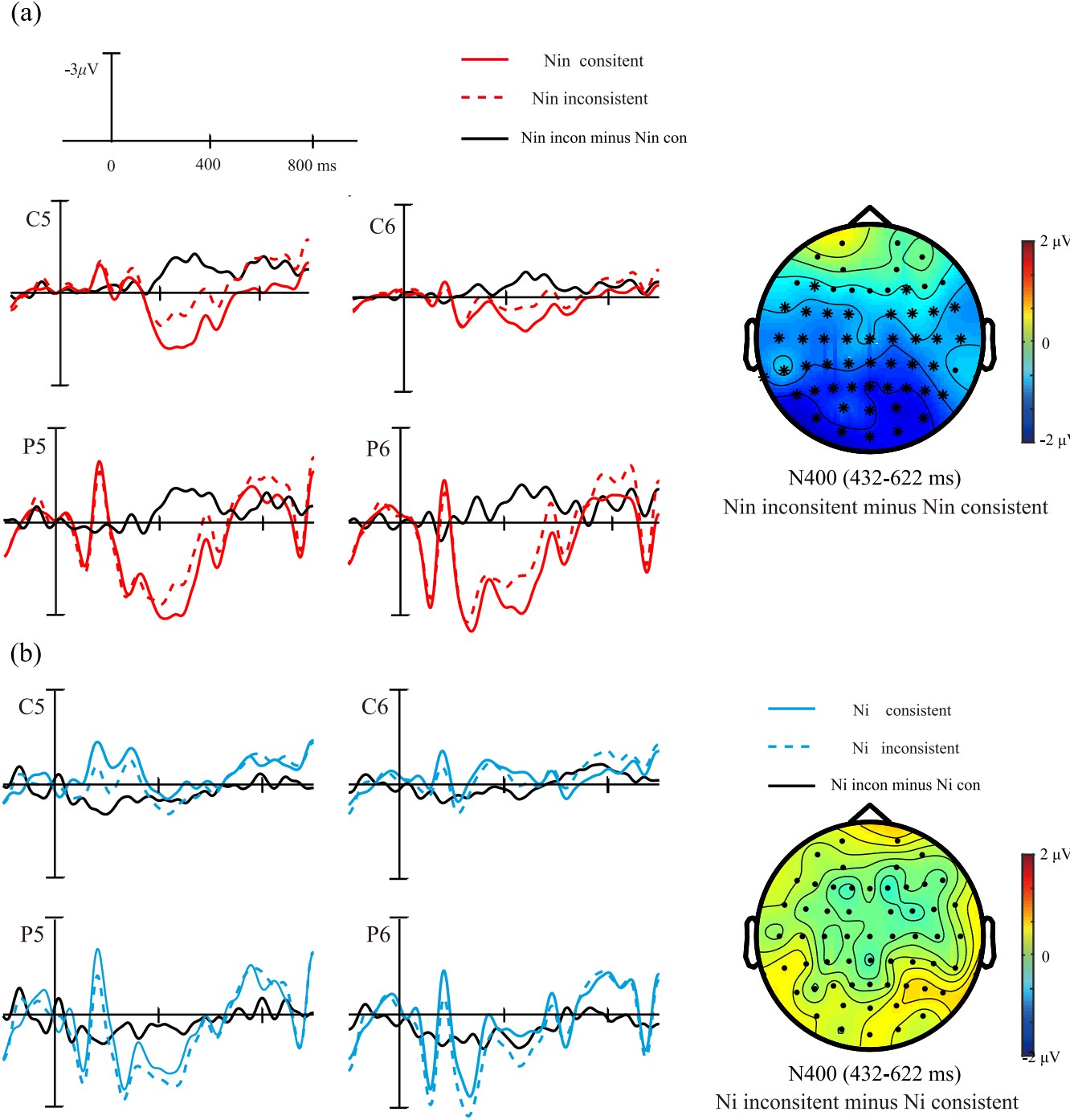

**Fig 2. The results of the cluster-based permutation tests to compare the consistency effect of different pronoun types.** (a) Difference waves and topography for Nin inconsistent minus Nin consistent. The asterisks in the topography indicate the electrodes that the difference between the two conditions reaches significance. (b) Difference waves and topography for Ni inconsistent minus Ni consistent.

However, no significant cluster was detected in the comparison between Ni inconsistent and Ni consistent ($ps > 0.1$).

## Discussion

To address the limitations of the previous study [7], we revisited the processing of different forms of second-person singular pronouns to further elucidate the neural basis of respectfulness processing in Mandarin Chinese. Traditional ANOVA analysis and cluster-based permutation tests were conducted on the data, and both analyses revealed similar ERP patterns. The purpose of the ANOVA analysis was to compare the results of the present study with those of Jiang et al.'s study [7]. Our ANOVA results showed a broadly distributed N400 and a late negativity response for the Nin inconsistent condition, but not for the Ni inconsistent condition. Unlike the findings of Jiang et al., which reported the N400 effect for both Nin and Ni inconsistent conditions, our findings demonstrated early distinct neural activities for these conditions during Chinese sentence comprehension. Furthermore, Jiang et al. [7] observed a sustained negative activity for the Ni inconsistent and sustained positive activity for the Nin inconsistent in the late time window, which reflects participants' strategies for resolving the inconsistency, varying according to different types of pronouns. In this study, we only observed late negativity for Nin inconsistent, not for Ni inconsistent. Consistent with our expectations, these results suggest that participants' strategies can be minimized by using various fillers and an implicit comprehension task. Notably, the cluster-based permutation tests showed similar results: a cluster of significant negativity for the Nin inconsistent compared to the Nin consistent condition was detected, extending from approximately 432 to 622 ms over the central and posterior regions. Additionally, no significant cluster was found between the Ni consistent and inconsistent conditions.

Compared with the conventional ANOVA analysis, the cluster-based permutation tests can provide more reliable ERP results because they considered all time points and electrodes. Thus, we believe that the N400 component and the late negative component reported in the ANOVA analysis should belong to one ERP component. In the present study and the study of Jiang et al. [7], there are insufficient reasons to subjectively divide the time window of 300–800 ms into two time windows of 300–500 ms and 500–800 ms because the pattern of ERP waveforms for different conditions is not clear cut. As shown in results from cluster-based permutation tests, Nin inconsistent elicited a larger negative activity compared to Nin consistent, whereas there was no significant difference between Ni inconsistent and consistent. The more negative cluster for Nin inconsistent compared to Nin consistent can be labeled as N400 because its topographical distribution was the same as the typical N400 response, despite the identified time window of N400 (432–622 ms) was a little later than that identified in previous studies (starting at ~300 ms). The ERPs result for Nin inconsistent is partly consistent with that for the different forms of verbs in Japanese [6]. Both Nin inconsistent condition in this study and inappropriate verbs in Japanese in the previous study [6] elicited larger N400 responses. This consistency suggests that the processing of honorific expressions is, partly, cross-linguistically common.

In a review on the context of language comprehension, Schumacher [12] proposed that the N400 seems to intrinsically be an index of a mismatch between the upcoming word and the prediction based on the context (the context including mutual knowledge, genre-specific properties, the identities of the speaker and the addressee, and the truth value of the sentence) [12,23–27, for a review see 12]. For example, the mismatch between the semantic meaning of the upcoming word and one's world knowledge (e.g., "the Dutch trains are white", actually, the trains in Dutch are yellow) elicited a larger N400 response than the match one (e.g., "the

Dutch trains are yellow") [26]. In the current study, the critical words in the Nin inconsistent condition constituted a violation of global social knowledge: a speaker of lower social status was expected to use the respectful form nin[respectful] with a listener of higher social status. Thus, the respectful form of the second-person pronoun "Nin" used by a speaker of higher social status for a listener of lower social status in the inconsistent condition was unexpected, resulting in a more pronounced N400 in the Nin inconsistent than in the consistent condition. Distinguished from the classic N400 response to semantic violation, the late latency of the current N400 response may reveal that the processing of the relative social status of the speaker and the addressee requires more inference, resulting in a delay in the processing of the second person pronoun.

The sustained late negativity that is related to inference intention found in the present study has been frequently reported in previous studies. This implies a second-pass process and computationally costly pragmatic inference [18,19,28–31]. A speaker of higher status used the respectful form of the second-person pronoun toward an addressee of lower status, which could have some specific purposes. For example, to ridicule the addressee, make jokes, or express a distant relationship with the addressee. An investigation from Mao [32] showed that, in addition to using the respectful form of the second person pronoun "Nin" to show respect, people use "Nin" to express ridicule (13%), make jokes (40%), or for serving other purposes (15%). In the present study, building a coherent interpretation requires one to construct a new meaning that goes beyond the lexical meaning of upcoming words. Therefore, the additional information that "the respectful form of the second-person pronoun can be used to emphasize ironic intention or make jokes" has been searched. This extra effort in elaborating the speaker's intention to yield contextually appropriate meaning could cause a larger negative activity. This explanation is consistent with that in Bambini et al.'s study [17]. They reported a similar ERP pattern for literary metaphors (an N400 followed by sustained negativity) to the results of ANOVA in the present study. They proposed that the negative activity for literary metaphors reflected an enduring effort in inferring the figurative meaning.

Finally, although the cluster-based permutation tests revealed that the difference between Nin inconsistent and consistent conditions extended from approximately 432 to 622 ms over the central and posterior regions, as shown by the waveforms and topographic distributions in Figs 1 and 2, it seems that this negative activity starts from an earlier time point and is sustained at a very late time point. Therefore, the N400 effect may be a Nref-like effect. The Nref effect is a sustained negative shift that has been reported in cases in which two antecedents are equally plausible referents for an anaphor [33,34], or when the definitional gender of the pronoun is inconsistent with the stereotypical gender of the antecedent [35,36], such as "herself" in the sentence "the architect saw herself in the mirror," relative to the "himself" [36]. In this study, readers must link the form of the second-person pronoun to the social status of the speaker in the prior context. The inconsistency between the form of the second-person pronoun and the prior context caused difficulty in the anaphora. Consistent with previous studies [36], this negative activity could be an index of more effort spent in bounding the pronoun "nin" to the inconsistent antecedent (a speaker of lower social status).

Meanwhile, the absence of an N400 effect and the late negativity for the Ni inconsistent condition could be due to the pragmatic property of $ni_{[plain]}$. Ni[plain] can be used in scenarios where there is a close relationship between a speaker of a lower status and an addressee of a higher status (e.g., between a daughter and her mother) [29,37,38]. Liang [38] studied the basic principles that govern the usage of $ni_{[plain]}$ and $nin_{[respectful]}$ and reported the use of $nin_{[respectful]}$ to show emotions other than respect, such as indignation and sarcasm. She also found people are more willing to address others with $ni_{[plain]}$ to show intimacy and familiarity. Therefore, $ni_{[plain]}$ violation may be more acceptable and would not cause difficulty in semantic meaning

integration. To examine this possibility, we asked 20 more participants (age range, 18–31 years; mean, 22.5 ± 4.75 years; 10 women) to rate the acceptability of the sentence segment from the critical materials (e.g., Professor Zhang said to student Lin: "your. . .") on a 5-point Likert scale (1 representing totally unacceptable and 5 representing totally acceptable). The results of paired sample *t* test showed that the rate score difference between Ni consistent and inconsistent (4.18 vs. 3.55) was significantly smaller than the difference between Nin consistent and inconsistent conditions (4.25 vs. 2.64, $p < 0.001$), indicating that readers perceived the Ni violation as more acceptable than the Nin violation.

The presence of an N400 for Ni inconsistent in Jiang et al. [7] might be due to methodological limitations, such as the baseline problem. As noted earlier, the context in that study was unbalanced across conditions. Moreover, Jiang et al. [7] reported a P200 effect: compared to $ni_{[plain]}$, an increased P200 response was elicited by $nin_{[respectful]}$. They attributed this difference to the more complex orthography of $nin_{[respectful]}$, but this effect may also result from the baseline spillover effect, which could further balance out the $nin_{[respectful]}$ effect, or exaggerate the $ni_{[plainl]}$ effect in the later time range. Additionally, Jiang et al. [7] found a sustained positivity in the 800–1600 ms time window to the Nin inconsistent condition. They interpreted the late positivity as being associated with a non-literal (sarcastic) interpretation of the anomalous $nin_{[respectful]}$. As noted in the introduction, the appearance of late positivity is influenced by the experimental task and environment [14,15,33]. Furthermore, according to the extended argument dependency model, a late positivity component could be elicited because of the well-formedness computation depending on the overall experimental task and environment [33]. Related to these findings, we questioned the explanation by Jiang et al. [7] (that the late positivity reflects a non-literal interpretation). Possibly, their late positivity for anomalous $nin_{[respectful]}$ could simply reflect an evaluation of the well-formedness of sentences that were specific to their experimental environment.

## Conclusions

In summary, we failed to replicate the main results of Jiang et al. [7]. Thus, our findings highlight the importance of various experimental materials in language studies. Future research should design and control experimental materials elaborately, such as the degree of plausibility of sentences, otherwise, the behavior patterns and neural activities underlying language comprehension are easily distorted. The current findings suggest that the Nin inconsistent condition causes difficulty in integrating the critical word with the prior context, reflected by an N400 response, whereas no neural difference is found between the Ni consistent and inconsistent conditions due to people being more likely to use the plain form $ni_{[plain]}$. Therefore, we believe that the differential neurocognitive processes underlying the representation of the appropriate form of the second-person pronoun according to the rank relationship of the interlocutors provide a novel insight into the linguistic processing of politeness.

## Supporting information

**S1 Table. Examples of filler sentences, and the number of sentences of each type.** (DOCX)

## Author Contributions

**Data curation:** Liyan Ji.

**Formal analysis:** Liyan Ji.

**Investigation:** Liyan Ji.

**Methodology:** Liyan Ji.

**Project administration:** Liyan Ji.

**Resources:** Liyan Ji.

**Software:** Liyan Ji.

**Supervision:** Liyan Ji.

**Validation:** Liyan Ji.

**Visualization:** Liyan Ji.

**Writing – original draft:** Liyan Ji.

**Writing – review & editing:** Liyan Ji, Lin Cai, Aiai Ji.

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
