## [Decision Letter · Decision Letter 0]

27 May 2021

PONE-D-21-12659

Respectfulness-processing revisited: An ERP study of Chinese sentence reading

PLOS ONE

Dear Dr. Ji,

Thank you for submitting your manuscript to PLOS ONE. After careful consideration, we feel that it has merit but does not fully meet PLOS ONE’s publication criteria as it currently stands. Therefore, we invite you to submit a revised version of the manuscript that addresses the points raised during the review process.

The Reviewers identified a number of points that should be clarified in your Manuscript. The literature is not properly covered and the implications of the present findings should be strengthened. Additional data could be added in an appendix. Finally, the statistical approach could be improved.

We look forward to receiving your revised manuscript.

Kind regards,

Nicola Molinaro, Ph.D.

Academic Editor

PLOS ONE

Journal Requirements:

3. Please change "female” or "male" to "woman” or "man" as appropriate, when used as a noun (see for instance https://apastyle.apa.org/style-grammar-guidelines/bias-free-language/gender).

6. Please ensure that you refer to Figure 4 in your text as, if accepted, production will need this reference to link the reader to the figure.

Reviewers' comments:

Reviewer's Responses to Questions

**Comments to the Author**

1. Is the manuscript technically sound, and do the data support the conclusions?

Reviewer #1: Partly

Reviewer #2: Yes

2. Has the statistical analysis been performed appropriately and rigorously? 

Reviewer #1: No

Reviewer #2: Yes

3. Have the authors made all data underlying the findings in their manuscript fully available?

Reviewer #1: No

Reviewer #2: No

4. Is the manuscript presented in an intelligible fashion and written in standard English?

Reviewer #1: No

Reviewer #2: No

5. Review Comments to the Author

Reviewer #1: This is an interesting study on pragmatic respectfulness effects on processing second person pronouns in Mandarin Chinese. The study seems appropriately motivated and described and well conducted. The results are fairly clear. I have some comments that the authors should consider when producing a new version of the paper:

· What are the (written and spoken) corpus frequencies of the two pronouns nin/ni? Can frequency effects explain the observed ERPs in the first time window (for inconsistent nin but not for inconsistent ni)?

· Please provide details about artifact rejection. How were artifacts identified? What counts as “excessive artifacts”?

· Please provide examples and a more detailed description of fillers. Please provide an analysis of ERP data for fillers (e.g., in appendix)

· Filtering (lines 202-203): be explicit about which filters were on-line (during recording) and which filters were applied off-line (during preprocessing)

· “Depending on the results of the baseline time window…”: this part was the least clear to me; please describe clearly how the baseline data were analysed and how results impacted the analysis of the main ERP comparisons

· On selecting windows for statistical analysis: I recommend using a clustering approach in the time/channel dimensions (e.g., Maris & Ostenveld 2007) to identify effects

· Some discussion about late negativities in ERP research is missing: several studies have reported such effects (e.g., Nref: Nieuwland & van Berkum 2008; SAN: Baggio et al. 2008 etc.), it would be important to discuss how the process of resolving respectfulness inconsistencies relates to other processes associated with similar ERP effects.

Reviewer #2: The study investigates the use of plain vs respectful second-person pronouns in Mandarin. The use of the pronoun is guided by politeness factors. It closely moves from the results of another work showing that the use of respectful-pro in an inconsistent context elicited an anterior negativity 300-500 and a sustained positivity 800-1600. the use of plain-pro in an inconsistent context instead elicited a broad negativity and a sustained negativity. Authors observe that the study has a few issues (lack of appropriate fillers and baseline problems), and want to assess the replicability of the findings. Differently from the study of Jiang et al 2013 authors report an asymmetry between pronoun types during the N400 time window: inconsistent (compared to consistent) use of the respectful-pro elicited a larger N400, which was not observed with plain-pro.

The study may be of interest, for PLOS readership, but its scope is rather narrow, considering that it closely moves from a previous work. Still, I think it has same merits in these times of replication crisis, but before being apt for publication I thinks authors should revise it, focusing on the writing quality and on the interpretation of the results, which may be more nuanced in several sections of the text. A few specific comments follow:

introduction

- perhaps the focus of the introduction could be more on the use of respectful forms of in chinese and on the reasons why it is an important topic. the analysis and critique of Jiang et al, should be a subsection of the introduction on its own. following the critique, authors should describe how "the present study" will overcome the previous study's limits and then clearly spell out the experimental predictions

method

- I cannot see anything wrong in the analysis of the ERP data. the only thing is that the details on the artifact rejection procedure should be moved in the ERP analysis section

- baseline correction is not explained clearly enough: it seems that instead of using 200 ms before pro, authors chose 200 ms before the onset of the whole sentence. if this is how it was done, it is fine for me, but it could be explained better.

discussion

- it is true that anterior negativities are quite often reported in the study of pragmatic phenomena. They are also found in literary metaphor comeprehension (Bambini, Canal, Resta & Grimaldi 2019 - Discourse processes), and when ambiguous anaphoric relations are computed (e.g., Nieuwland 2014, Neuropsychologia ; Canal, Garnham & Oakhill, 2015, Frontiers). In metaphor it has been interpreted as reflecting mechanisms related to the drawing of an array of weak implicatures, rather than one straightforward implicature (which is instead associated with larger P600/LPC). in anaphor procesing the Nref is associated with the search for additional information in the mental representation of the discourse.

These studies may help the authors in elaborating a little bit more on the functional meaning of their findings, which as it is now has very limited implications. Also discourse linking and update mechanisms proposed by Petra Schumacher may help in the interpretation of the N400 effect in the context of pronoun resolution (Schumacher, P. B. (2012). Context in neurolinguistics. In R. Finkbeiner, J. Meibauer, & P. B. Schumacher (Eds.), What is a Context?: Linguistic Approaches and Challenges (pp. 33–53).

There are many typos, and the writing quality should be improved

- to make data fully available they should be provided as part of the manuscript or its supporting information, or deposited to a public repository.

6. PLOS authors have the option to publish the peer review history of their article (what does this mean?). If published, this will include your full peer review and any attached files.

Reviewer #1: No

Reviewer #2: **Yes: **Paolo Canal

---

## [Author Response · Author response to Decision Letter 0]

21 Jul 2021

We thank the reviewers for their constructive suggestions concerning revisions to the manuscript. The revised parts of the main text are highlighted in blue. We detailed our responses to the reviewers’ points below.

Reviewers' comments:

Review Comments to the Author

Reviewer #1: This is an interesting study on pragmatic respectfulness effects on processing second person pronouns in Mandarin Chinese. The study seems appropriately motivated and described and well conducted. The results are fairly clear. I have some comments that the authors should consider when producing a new version of the paper:

1. What are the (written and spoken) corpus frequencies of the two pronouns nin/ni? Can frequency effects explain the observed ERPs in the first time window (for inconsistent nin but not for inconsistent ni)?

Response: The present study reported a more negative response for Nin inconsistent than Nin consistent, but not for the comparison between Ni inconsistent and Ni consistent. These results can not be due to the frequency difference of the two pronouns. First, the usage frequency for nin is 515 per million and 39629 per million for ni (Cai and Brysbaert, 2010). Generally, words with a word frequency greater than 50 per million are considered high frequency words, so both critical words (ni and nin) are high frequency words (Wang and Zhang, 2021). Furthermore, we didn’t compare the pronoun Nin with Ni directly in the present study, but compared the effect for Nin consistency (the difference between Nin inconsistent and Nin consistent) with the effect for Ni consistency (the difference between Ni inconsistent and Ni consistent). Therefore, the frequency effects can not explain the observed ERPs in the present study.

We have added this information on page 9 (lines 170-171) as follows:

 “ There is a high-frequency usage for both nin (515 per million) and ni (39629 per million) [20].”

2. Please provide details about artifact rejection. How were artifacts identified? What counts as “excessive artifacts”?

Response: The obvious artifacts (over 150 μV) were first deleted by visual inspection before the data analysis. After epoching, trials with a voltage, relative to the 200 ms baseline, exceeding ±65μV at any electrode were excluded through artifact rejection. Participants with less than 30 trials for each condition left were identified as participants with excessive artifacts and were excluded from further data analysis.

We have supplemented these details in the EEG data Preprocessing section on page 12, lines 237-244.

3. Please provide examples and a more detailed description of fillers. Please provide an analysis of ERP data for fillers (e.g., in appendix)

Response: We apologize for the unclear descriptions of fillers. We have added a table of example sentences of fillers in supplementary materials (see Table S1), and provided a more detailed description of fillers on page 10. line 186-189.

We didn’t conduct ERP analysis on the ERP data of fillers since the critical word in the fillers did not counterbalanced. Furthermore, the result of fillers won’t help for the explanation of the present results.

4. Filtering (lines 202-203): be explicit about which filters were on-line (during recording) and which filters were applied off-line (during preprocessing).

Response: We apologize for the unclear descriptions. We have added the information about filtering on page 11(lines 222-224) as follows:

“The EEG signals were sampled at 500 Hz with a band-pass filtered 0.016–100 Hz online and were filtered again offline with a band-pass of 0.1–40 Hz for data analysis. ”

5. “Depending on the results of the baseline time window…”: this part was the least clear to me; please describe clearly how the baseline data were analysed and how results impacted the analysis of the main ERP comparisons.

Response: We apologize for the unclear descriptions. In order to rule out the potential influence of prior context on the ERP effect for the critical word, we checked baseline differences by using the blank interval prior to the presentation of the first word as the baseline. A long epoch was conducted from 200 ms prior to the first word and 3200 ms before presenting the critical word, then the time window of 3000-3200 ms was the baseline for the critical word analysis (see figure in "response to reviewers" for illustration). Statistical analyses were conducted on the average amplitude during the 3000-3200ms time window. The result showed that there was no significant main effect or interaction involving factors of pronoun type and pronoun consistency. Thus, the potential baseline problem could be excluded and then baseline correction could be conducted in the following analyses. Otherwise, baseline correction would not be conducted and instead a high pass filtering would be applied as suggested by Widmann et al. (2014).

We have rewritten the description of baseline correction in the revised manuscript (page 12, lines 230-234).

6. On selecting windows for statistical analysis: I recommend using a clustering approach in the time/channel dimensions (e.g., Maris & Ostenveld 2007) to identify effects

Response: Thanks for your suggestions on the statistical methods. We have conducted the cluster-based permutation tests, and added the details of these tests and the results in the updated manuscript (pages 13-14, page 17).

7. Some discussion about late negativities in ERP research is missing: several studies have reported such effects (e.g., Nref: Nieuwland & van Berkum 2008; SAN: Baggio et al. 2008 etc.), it would be important to discuss how the process of resolving respectfulness inconsistencies relates to other processes associated with similar ERP effects.

Response: We have reorganized the Discussion section and incorporated related studies in the revision (pages 18-22). 

Reviewer #2: The study investigates the use of plain vs respectful second-person pronouns in Mandarin. The use of the pronoun is guided by politeness factors. It closely moves from the results of another work showing that the use of respectful-pro in an inconsistent context elicited an anterior negativity 300-500 and a sustained positivity 800-1600. the use of plain-pro in an inconsistent context instead elicited a broad negativity and a sustained negativity. Authors observe that the study has a few issues (lack of appropriate fillers and baseline problems), and want to assess the replicability of the findings. Differently from the study of Jiang et al 2013 authors report an asymmetry between pronoun types during the N400 time window: inconsistent (compared to consistent) use of the respectful-pro elicited a larger N400, which was not observed with plain-pro.

The study may be of interest, for PLOS readership, but its scope is rather narrow, considering that it closely moves from a previous work. Still, I think it has same merits in these times of replication crisis, but before being apt for publication I thinks authors should revise it, focusing on the writing quality and on the interpretation of the results, which may be more nuanced in several sections of the text. A few specific comments follow:

introduction

1. perhaps the focus of the introduction could be more on the use of respectful forms of in chinese and on the reasons why it is an important topic. the analysis and critique of Jiang et al, should be a subsection of the introduction on its own. following the critique, authors should describe how "the present study" will overcome the previous study's limits and then clearly spell out the experimental predictions

Response: We have reorganized the introduction section in the updated manuscript.

method

2. I cannot see anything wrong in the analysis of the ERP data. the only thing is that the details on the artifact rejection procedure should be moved in the ERP analysis section

Response: We have moved the details on the artifact rejection to the section of EEG data preprocessing on page 12, lines 230-237.

3. baseline correction is not explained clearly enough: it seems that instead of using 200 ms before pro, authors chose 200 ms before the onset of the whole sentence. if this is how it was done, it is fine for me, but it could be explained better.

Response: We apologize for this unclear description. In the present study, we used the 200ms before the critical word as a baseline. In order to rule out the potential influence of prior context on the ERP effect for the critical word, we first checked baseline differences by using the blank interval prior to the presentation of the first word as the baseline. Then a long epoch was conducted from 200 ms prior to the first word and 3200 ms before presenting the critical word (time-locked to the first word), thus the time window of 3000-3200 ms was the baseline for the critical word analysis (see figure in "response to reviewers" for illustration). Statistical analyses were conducted on the average amplitude during the 3000-3200ms time window. The result showed that there was no significant main effect or interaction involving factors of pronoun type and pronoun consistency. Therefore, the potential baseline problem could be excluded, and it is safe to use the 200-ms before the critical word as a baseline, then baseline correction was conducted on the analysis of the epoch time locked to the critical word.

We have rewritten the description of baseline correction in the revised manuscript (page 12, lines 230-234).

discussion

4. it is true that anterior negativities are quite often reported in the study of pragmatic phenomena. They are also found in literary metaphor comeprehension (Bambini, Canal, Resta & Grimaldi 2019 - Discourse processes), and when ambiguous anaphoric relations are computed (e.g., Nieuwland 2014, Neuropsychologia ; Canal, Garnham & Oakhill, 2015, Frontiers). In metaphor it has been interpreted as reflecting mechanisms related to the drawing of an array of weak implicatures, rather than one straightforward implicature (which is instead associated with larger P600/LPC). in anaphor procesing the Nref is associated with the search for additional information in the mental representation of the discourse.

These studies may help the authors in elaborating a little bit more on the functional meaning of their findings, which as it is now has very limited implications. Also discourse linking and update mechanisms proposed by Petra Schumacher may help in the interpretation of the N400 effect in the context of pronoun resolution (Schumacher, P. B. (2012). Context in neurolinguistics. In R. Finkbeiner, J. Meibauer, & P. B. Schumacher (Eds.), What is a Context?: Linguistic Approaches and Challenges (pp. 33–53).

Response: Thank you very much for these helpful suggestions. We have rewritten the section of Discussion and incorporated these related studies in the Discussion section on pages 18-22.

5. There are many typos, and the writing quality should be improved

Response: The revised manuscript has been checked by professional language editing services.

6. to make data fully available they should be provided as part of the manuscript or its supporting information, or deposited to a public repository. 

Response: The original data and relevant supporting information in the present study are available at https://osf.io/zf35p

---

## [Editor Report · Decision Letter 1]

18 Aug 2021

PONE-D-21-12659R1

Respectfulness-processing revisited: An ERP study of Chinese sentence reading

PLOS ONE

Dear Dr. Ji,

Thank you for submitting your manuscript to PLOS ONE. After careful consideration, we feel that it has merit but does not fully meet PLOS ONE’s publication criteria as it currently stands. Therefore, we invite you to submit a revised version of the manuscript that addresses the points raised during the review process.

While superficially reading the Manuscript I noticed a couple of typos:

- lines 232-233: - A long epoch was "conducted" from 200 ms before the onset - I would write - A long epoch was "computed" from 200 ms before the onset -

- line 401:  - resulting in a delay the processing of the second - I would write - resulting in a delay "of" the processing of the second -

I am wondering if other typos are present and I invite the authors to carefully revise the whole Manuscript before final acceptance.

A marked-up copy of your manuscript that highlights changes made to the original version. You should upload this as a separate file labeled 'Revised Manuscript with Track Changes'.An unmarked version of your revised paper without tracked changes. You should upload this as a separate file labeled 'Manuscript'.

We look forward to receiving your revised manuscript.

Kind regards,

Nicola Molinaro, Ph.D.

Academic Editor

PLOS ONE
---

## [Author Response · Author response to Decision Letter 1]

28 Sep 2021

Review Comments to the Author

Thank you for submitting your manuscript to PLOS ONE. After careful consideration, we feel that it has merit but does not fully meet PLOS ONE’s publication criteria as it currently stands. Therefore, we invite you to submit a revised version of the manuscript that addresses the points raised during the review process.

While superficially reading the Manuscript I noticed a couple of typos:

- lines 232-233: - A long epoch was "conducted" from 200 ms before the onset - I would write - A long epoch was "computed" from 200 ms before the onset -

- line 401: - resulting in a delay the processing of the second - I would write - resulting in a delay "of" the processing of the second -

I am wondering if other typos are present and I invite the authors to carefully revise the whole Manuscript before final acceptance.

Response: Thank you very much for your careful reading. We have double checked the typos and grammar of the manuscript, and the revised parts of the manuscript are highlighted in red.

---

## [Editor Report · Decision Letter 2]

1 Oct 2021

Respectfulness-processing revisited: An ERP study of Chinese sentence reading

PONE-D-21-12659R2

Dear Dr. Ji,

We’re pleased to inform you that your manuscript has been judged scientifically suitable for publication and will be formally accepted for publication once it meets all outstanding technical requirements.

Kind regards,

Marte Otten, Ph.D.

Academic Editor

PLOS ONE
---

## [Editor Report · Acceptance letter]

16 Jun 2022

PONE-D-21-12659R2 

Respectfulness-processing revisited: An ERP study of Chinese sentence reading 

Dear Dr. Ji:

I'm pleased to inform you that your manuscript has been deemed suitable for publication in PLOS ONE. Congratulations! Your manuscript is now with our production department. 

Kind regards, 

on behalf of

Dr. Marte Otten 

Academic Editor

PLOS ONE